# The Calibration Methods of Geometric Parameters of Crystal for Mid-Infrared Acousto-Optic Tunable Filter-Based Imaging Systems Design

**DOI:** 10.3390/ma16062341

**Published:** 2023-03-14

**Authors:** Kai Yu, Qi Guo, Huijie Zhao, Chi Cheng

**Affiliations:** 1School of Instrumentation Science & Opto-Electronics Engineering, Beihang University, No. 37 Xueyuan Road, Haidian District, Beijing 100191, China; 2Key Laboratory of “Precision Opto-Mechatronics Technology”, Ministry of Education, No. 37 Xueyuan Road, Haidian District, Beijing 100191, China

**Keywords:** AOTF spectral imager, geometric calibration of AO crystal, spectral response, imaging aberrations, tuning curve

## Abstract

AOTF calibration is a complex topic that has various aspects. As far as geometric calibration is concerned, it includes not only processing errors and fixing errors in the optical system, but also the error of geometric parameters of crystal (GPC). GPC is the preset input in the optical design and optimization of Zemax, which determines the key parameters, including the spatial resolution, the field of view, and aberration. In particular, the compensation of aberration during the optical design requires accurate values of GPC. However, it is currently considered ideal. Therefore, two calibration methods based on the principle of parallel tangent are proposed: (1) the minimum-central wavelength method; (2) the minimum-frequency method. The deviation of the parallel tangent incident angle calibrated by the two methods is 0.03°. As a result, the tuning curve calculated in theory with the calibrated geometric parameters of AOTF is consistent with the tuning curve measured in practice.

## 1. Introduction

The acousto-optic tunable filter (AOTF) has been widely applied in spectral imaging systems for remote sensing [1,2], spectral microscopy [3,4], three-dimensional detection [5], notch filtering [6,7], and edge enhancement [8] due to its advantages of a narrow bandpass, electric tunability, all-solid-state, fast response, large angular aperture, and wide spectral range. In particular, the flexible electric tunability makes AOTF spectral imagers the most promising candidates for identifying and tracking targets. The central wavelength can be arbitrarily changed by modulating the frequency of the acousto-optic grating generated in a birefringent crystal material with an ultrasonic transducer. Until now, novel structures of AOTF imaging spectrometers have been developed for the purpose of performance enhancement—for instance, to improve the resolution of the spectrum or of space [9,10]. Moreover, some new crystalline materials [11] have emerged in the field of AOTF devices to improve the spectral range and efficiency. The low price and high spectral resolution of AOTF indicate its potential application in the field of plasmonic sensing [12].

The calibration of AOTF is inseparable from the study of device characteristics and system design. In the area of modeling for AOTF devices, Zhang was the first to propose a noncollinear AOTF operating mode [13] and a design scheme according to the principle of parallel tangent [14]. This principle is the criterion for the design of a wide-aperture AOTF, which needs to be considered before designing. Yushkov [15,16] put forward the analytical expressions of the central wavelength and tuning frequency at oblique incidence. The spectral response of AOTF depends not only on the GPC but also on the direction of incident light. Before the AOTF spectrometer is processed, the imaging characteristics need to be simulated by modeling. In traditional methods of the optical system design of an AOTF spectrometer, the AOTF device is considered as a transparent slab, which is far from sufficient in terms of aberration evaluation and optical design optimization. Batshev and Machikhin [17] summarized and tested four typical optical schemes in common use: confocal, collimating, tandem, and double-path. As a special component in an optical system, AOTF cannot be represented by the existing surface of Zemax. The method of using the user-defined surface (UDS) function in Zemax and computer dynamic link library (DLL) technique to design an optical system was introduced [18]. The GPC is the input of UDS. Subsequently, Machikhin solved the limitation of the field of view of AOTF with a computational technique [19]. It can be found the device characteristics of AOTF are crucial for system optimization, and GPC directly affects the device characteristics. However, little attention was paid to the accuracy of GPC.

AOTF needs to be calibrated in practical applications. Calibrations mainly include spectral, geometric, radiometric, thermal, etc. For spectral calibration, it is necessary to establish the mapping relationship between spectrum and frequency [20], which is usually characterized by the tuning curve [21,22]. For geometric calibration, scholars are mostly committed to the research of system-level calibration methods. For instance, Pozhar, Machikhin [23,24,25], and Liu Hong [26] focused on the quantitative description of the AOTF spatial aberrations. As mentioned earlier, the aberrations of the system are caused by the coupling of many factors. The influence of GPC is not considered separately. Shi [27] proposed a multiplane camera calibration model (MPM). Together with the fringe-phase marking method and the neural network algorithm, MPM established the mapping between the image points and the corresponding space lines. Katrašnik [28,29] proposed some fully automatic geometrical calibration methods for the hyper-spectral imaging systems based on AOTF. However, they could only aim at imaging samples on a 2D plane due to the defocusing of 3D objects within a broad spectral range. For radiometric calibration, the accuracy of the radiometric response of AOTF hyperspectral imaging systems is crucial for obtaining reliable measurements. Katrašnik analyzed the influence of the noise of the detector on the radiometric calibration accuracy [30]. The charge-coupled device (CCD) imaging sensor was thoroughly analyzed in [31], while the complementary metal–oxide–semiconductor (CMOS) imaging sensor was analyzed in [32]. The difference in spectral value caused by the inhomogeneity of temperature in the crystal, such as anomalous sidelobes [33], and the drift of the central wavelength, was often considered in the thermal calibration [34]. As a part of AOTF calibration, the geometric calibration of the system is deservedly important, but it cannot guide the optimization of an optical system. The geometric aberrations are the result of the joint influence of the mirror group and AOTF in the optical system. The vertical incidence angle ($\theta_{i}$) and ultrasonic cut angle (α) determine the GPC, which ultimately affects the performance of the system. The influence of the wedge angle of the AOTF rear surface in the mid-infrared band is very small and generally not considered. Normally, most devices follow the design rules proposed by Zhang, where $\theta_{i}$ and α are constrained under the principle of parallel tangent [13]. In this case, $\theta_{i}$ can be calculated once α is fixed. However, for devices not designed under this principle, there is no effective method to obtain α and $\theta_{i}$ accurately. In addition, since the GPC may shift from the designed values due to the sophistication during the fabrication process of AOTF devices, a feasible and accurate calibration of GPC is necessary.

Table 1 summarizes the works related to AOTF in recent years. The table includes calibration, optical design and optimization, operating modes, and ray tracing.

The different GPC directly affect the distribution of relative diffraction efficiency, spectral response, and frequency response of the device. On the contrary, we can use the characteristics of the test results to obtain the desired GPC. Therefore, we propose two methods: (1) the minimum-central wavelength method; (2) the minimum-frequency method. The calibration of GPC should be completed before the design and optimization of the optical system based on AOTF. Ultimately, the accuracy of the simulation of spectral resolution, imaging distortion, spectral bandwidth, and other performance indexes can be guaranteed. The proposed methods can also be used in a polarizer-free AOTF-based spectral imaging system [35], as well as the installation and adjustment of AOTF imaging systems.

## 2. Background

At present, TeO_2_ is widely used as an acousto-optic material in the spectral span from visible light to mid-infrared. For non-collinear anisotropic acousto-optic interaction in TeO_2_, there are two working modes. The arbitrarily polarized incident beam is split into two orthogonally polarized propagation modes: the extraordinary “e” mode and the ordinary “o” mode. We can limit the polarization direction of incident light by adding a polarizer in front of the AOTF. For instance, when the incident light is extraordinary, diffracted light will be shifted to normal light. This working mode can be recorded as e→o mode. Correspondingly, the other working mode is o→e mode.

The main equations describing AO interaction may be derived from the laws of the conservation of energy and momentum for photons and a phonon [36]:(1)$$\{\begin{array}{c} \omega_{d}=\omega_{i}\pm f\\ K_{d}=K_{i}\pm K_{a} \end{array},$$where $\left(f,K_{a}\right)$,$\left(\omega_{i},K_{i}\right)$, and $\left(\omega_{d},K_{d}\right)$ are the frequencies and wavevectors of sound, incident light, and diffracted light. The sign “+” corresponds to the absorption of the phonon, and the sign “−” corresponds to its stimulated birth. $\left|K_{i}\right|=2\pi n_{i}(\lambda )/\lambda$, $\left|K_{d}\right|=2\pi n_{d}(\lambda )/\lambda$, $\left|K_{a}\right|=2\pi f_{a}/V_{a}$, $n_{i}(\lambda )$, and $n_{d}(\lambda )$ represent the refractive index of incident and diffracted light, respectively; $f_{a}$ and $V_{a}$ represent the ultrasonic frequency and acoustic phase velocity, respectively. The actual device will adjust the power of the ultrasonic transducer at different incident wavelengths to achieve high diffraction efficiency in each wave band.

Spectral imaging applications require that the Bragg phase matching condition $K_{d}=K_{i}+K_{a}$ for the wave vectors of ultrasound $K_{a}$, incident $K_{i}$, and diffracted $K_{d}$ light must be approximately satisfied in a wide range of incident light angles. This wide-aperture configuration of acousto-optic (AO) interaction [13] is achievable when the tangents to the wave surfaces for incident and diffracted light are parallel (Figure 1).

In Figure 1, λ is the wavelength of incident light. $n_{o}(\lambda )$ and $n_{e}(\lambda )$ represent the main refractive index of ordinary light and extraordinary light in the crystal. $\theta_{i}$ is the vertical incidence angle; α is the ultrasonic cut angle. The length of vectors $K_{ei}$ and $K_{od}$ depends on the refractive indices $n_{o}(\lambda )$ and $n_{e}(\lambda )$; the wavenumber $k_{0}$ corresponds to the light wavelength in a vacuum: $k_{0}=2\pi /\lambda$. AB is parallel to CD under the constraint of the principle of parallel tangent. In this case, the incident light angle is the parallel tangent incident angle ($\theta_{i}(\alpha )$). δθ is the offset between $\theta_{i}$ and $\theta_{i}(\alpha )$. According to the equations for calculating the refractive index of ordinary light and extraordinary light in anisotropic crystal, the refractive index of the refracted light in the crystal can be obtained once the wavelength of the incident light is determined, as shown in Equation (2) [9]. In addition, the geometric relationship can be deduced—that is, Equation (3) needs to be satisfied at the same time. In the mid-infrared band, Berny and Georgiev proposed the following dispersion equations and coefficients [37,38].
(2)$$\left\{ \begin{array}{l} n_{i}={\left(\cos^{2}\theta_{i}(\alpha )/n_{o}^{2}(\lambda )+\sin^{2}\theta_{i}(\alpha )/n_{e}^{2}(\lambda )\right)}^{-1/2}\\ n_{d}=n_{o}(\lambda ) \end{array}\right.,$$
(3)$$\{\begin{array}{c} \tan\theta_{d}={\left(n_{o}(\lambda )/n_{e}(\lambda )\right)}^{2}\tan\theta_{i}(\alpha )\\ \tan\left(\alpha \right)=\left(n_{d}\cos\theta_{d}-n_{i}\cos\theta_{i}(\alpha )\right)/\left(n_{i}\sin\theta_{i}(\alpha )-n_{d}\sin\theta_{d}\right) \end{array},$$

Once the wavelength of incident light is determined, α and $\theta_{i}(\alpha )$ will present a one-to-one correspondence. At the same time, the value of the ultrasonic cut angle α is also limited. $\theta_{i}(\alpha )$ is 55.53° when α is equal to 18.9°. If α continues to increase, the incident light angle that meets the constraint of the principle will not be found in the acousto-optic plane, losing practical significance. Therefore, the value range of α should be considered in the cutting design of AOTF. 

In the past, for AOTF with wide-aperture geometry, the tuning curve was mostly used to correct the GPC. When the wavelength of incident light is determined, the length of the acoustic vector is only related to the GPC, and the matching frequency $f_{a}$ of the driver can be obtained eventually, which can be expressed as follows [21]:(4)$$f_{a}=\frac{\left|K_{a}\right|V_{a}}{2\pi }=F(\lambda ,\alpha ,\theta_{i}),$$
where $V_{a}$ is the acoustic phase velocity, and $f_{a}$ is the acoustic frequency. However, Equation (4) is workable only if the default condition ($\theta_{i}=\theta_{i}(\alpha )$) is met. Then, the modified ultrasonic cut angle can be obtained by using Equation (5) [21].
(5)$$\alpha_{m}=\arg\underset{\alpha }{\min}\sum_{i=1}^{M}{\left|f_{ai}-f_{mi}\right|}^{2},$$
where $f_{ai}$ is the matching frequency calculated theoretically. $f_{mi}$ is the measured matching frequency. $\alpha_{m}$ is the modified ultrasonic cut angle. Generally, a small correction to α can make the theoretically calculated tuning curve very close to the measured tuning curve. If the difference between the corrected value of α and the nominal value of α given by the manufacturer is greater than 0.1°, the device does not strictly meet the parallel tangent principle, and the GPC needs to be calibrated separately. Of course, we can continue optimizing the value of α to match the tuning curve, but the optimization results at this time are not the real geometric parameters. The large deviation of α will directly affect the simulation of the spectral resolution, spatial resolution, aberrations, and other indicators, also bringing large errors to the optical simulation and design.

## 3. Simulation and Method

The mid-infrared AOTF is used as the simulation object, and the ray tracing method is a three-surface model that we proposed in our previous work [39]. This model can calculate the spectral response and frequency response at different incident light angles by numerical solution. The simulated optical scheme of the system is shown in Figure 2.

A parallel light structure is applied to the AOTF imaging spectrometer, the incident light is polychromatic, the driving frequency of AOTF is set to be 13.27 MHz, and the corresponding central wavelength of diffracted light is 4000 nm, calculated via the tuning curve. The column direction of the image plane corresponds to the direction of acousto-optic diffraction (within the color plane in Figure 3a). The central wavelength drift of the image plane can be obtained through simulation; the result is shown in Figure 3a. Specifically, when the frequency of the driver is fixed, the wavelength of diffracted light will shift with the change in the angle of incident light. Similarly, if the wavelength of incident light is fixed at 4000 nm, the central frequency drift of the image plane can be obtained, as shown in Figure 3b.

Here, we take the central wavelength shift as an example. Different positions of the image plane correspond to different incident light angles. From Figure 3a, it can be seen that the central wavelength drift on the image plane is symmetrically distributed in the row direction. In the column direction, the drift on one side is more serious than that on the other side. The wavelength of diffracted light reaches the lowest point ideally when the angle of incident light is perpendicular to the front surface of the crystal, which means that the central wavelength of the momentum matching point reaches a minimum when the angle of incidence satisfies the principle of parallel tangent ($\theta_{i}=\theta_{i}(\alpha )$). If the GPC does not meet this principle, the spectral response will change. Assuming that the vertical incidence angle ($\theta_{i}$) deviates from the parallel tangent incidence angle ($\theta_{i}(\alpha )$) by ±1°, the simulation results are as shown in Figure 4.

In Figure 4, the polar angle of incident light ($\theta_{p}$) within the crystal represents the angle between the incident light within the crystal and the optical axis. $\theta_{i}(\alpha )$ corresponding to the three GPCs remains unchanged and can be calculated from α directly. Therefore, in turn, an accurate GPC can be obtained through the angle difference ($\delta \theta_{1}$ or $\delta \theta_{2}$) between $\theta_{i}(\alpha )$ and $\theta_{i}$. For the AOTF device with undefined design parameters, accurate parameters can be obtained by seeking the lowest point of matching central wavelength or central frequency. The differences between the calibration optical scheme of the device and the optical scheme of the system lie in the following aspects: (1) because of the asymmetry of spectral response in the direction of acousto-optic diffraction, it is necessary to accurately change the angle of incident light in this direction; (2) the radiance of the infrared laser is too strong, so an optical power meter should be used to measure the diffraction light intensity instead of a detector; (3) during the calibration process, the position of the diffracted light will follow the change in the incidence angle and the wavelength (or the driving frequency), which cannot be obtained accurately. Moreover, the receiving aperture of an optical power meter is limited, so the method of measuring the radiance of the non-diffracting light is used to obtain the intensity of the diffracted light indirectly. It is noteworthy that the angle processing accuracy of AOTF is ±1°; therefore, an order of magnitude higher is needed for the accuracy of our angle control so as to meet the calibration requirements. The angle of the incident light is not easy to control accurately, so a precise turntable is used to change the angle of the AOTF device instead. The schematic diagram of the AOTF geometric parameter calibration is shown in Figure 5.

The AOTF is placed on a precision turntable, and the accuracy of the angle adjustment of the turntable is 0.01°. The light produced by the laser is elliptically polarized. A polarizer needs to be added before AOTF to ensure that the incident light is extraordinary. The power of zero-order light in the on and off states of the AOTF is obtained from the optical power meter, and the difference between the two is the power of the diffracted light. The deflection angle of the AOTF is positive when the AOTF rotates clockwise. At the same time, it is equal to the absolute value of the polar angle of the incident light, the sign opposite (the deflection direction is opposite). Here, the incident polar angle refers to the angle between the incident light outside the crystal and the optical axis.

The procedure of calibration of GPC is as follows:Ensure that the incident laser is perpendicular to the front surface of the AOTF;Adjust the rotation angle of the AOTF using the precision turntable;The spectral response at different angles can be obtained by adjusting the output wavelength of the laser (the minimum-central wavelength method);The frequency response at different angles can be obtained by adjusting the frequency of the transducer (the minimum-frequency method).

To improve the fitting accuracy of the final data, the angle step value is set as 1° for the large deflection angle of the AOTF, while the angle step value is selected as 0.1° when it is close to the parallel tangent incidence angle. Because the temperature of the AOTF has a certain influence on the spectral response, to shorten the acquisition time and to recover the spectra as completely as possible, the wavelength scan interval is set to be 10 nm when the diffraction efficiency is lower than 20%, 5 nm when the diffraction efficiency is 20~50%, and 1 nm when the diffraction efficiency is higher than 50%. During the test, the temperature change range of the crystal is controlled within ±0.1° by temperature control equipment.

## 4. Experiment and Results

### 4.1. The Minimum-Central Wavelength Method

According to the experimental setup in Figure 5, the frequency of the AOTF remains fixed at 14.0 MHz, and the wavelength of the laser needs to be tuned. The wavelength tuning range of the laser is 3674.0~4566.7 nm.

The test data of each spectral response are fitted with the least squares method to find the central wavelength corresponding to the point with the highest diffraction efficiency, as shown in Equation (6).
(6)$$\{\begin{array}{c} DE_{ci}=x(1)\times \sin c{\left(x(2)\times (\lambda_{mi}-x(3))\right)}^{2}+x(4)\\ L(DE_{mi},DE_{ci})=\sum_{i=1}^{m}{\left(DE_{mi}-DE_{ci}\right)}^{2} \end{array},$$
where x is a matrix of 1 × 4, which represents the initial value of the least squares method. $\lambda_{mi}$ represents the tested wavelength. $DE_{ci}$ represents the calculated diffraction efficiency and $DE_{mi}$ represents the tested diffraction efficiency. We minimize $L(DE_{mi},DE_{ci})$ by iteration, and x is the fitting coefficient.

Spectral responses at some incidence angles are shown in Figure 6. Figure 6a,b represent spectral responses at different incident polar angles. Different incident polar angles indicate that the angle of incident light changes in the acousto-optic interaction plane. The angle of incident light can be accurately obtained from the precision turntable. The central wavelength of the matched curve is the extreme value of the fitted curve, in order to reduce the error caused by random sampling.

Figure 7 shows the results of the minimum-central wavelength method. Figure 7a is a three-dimensional plot of the spectral responses at different angles. Figure 7b is the fitting curve of the central wavelengths; $\theta_{p}$ corresponding to the lowest point of the central wavelengths is 0.49°, and the corresponding central wavelength is 3960.6 nm. At this time, the angle of incident light is not perpendicular to the front surface of the AOTF. Thus, it can be seen that the principle of parallel tangent is not satisfied for this mid-infrared AOTF device, even presenting a large deviation.

Here, $\theta_{p}$ corresponding to the minimum central wavelength is marked as θ′. The offset between $\theta_{i}$ and $\theta_{i}(\alpha )$, i.e., δθ, can be obtained by the refraction law, as shown in Equation (7).
(7)$$\sin(\theta \prime )=n_{e}(\lambda_{s})\sin(\delta \theta ),$$
where $\lambda_{s}$ is the wavelength of the laser. In this experiment, the incident polar angle is 0.49°, $\delta \theta =0.22^{\circ }$ accordingly. According to the actual position of the transducer and the geometric relationship in the coordinate system, $\theta_{i}$ satisfies Equation (8).
(8)$$\theta_{i}=\theta_{i}(\alpha )+\delta \theta ,$$

### 4.2. The Minimum-Frequency Method

The difference between the two methods is that the fixed parameter in the minimum-frequency method is the output wavelength of the laser, while, in the minimum-central wavelength method, the frequency of the RF is relatively fixed. As shown in Figure 8a, $f_{1}$, $f_{2}$, and $f_{3}$ indicate the selection of three ultrasound frequencies. There will be two intersections with the spectral drift curve when the frequency is selected as $f_{1}$. The two points meet the momentum-matching condition and the diffraction efficiency is high. However, because the spectral drift in acousto-optic planes has no symmetry characteristic, the incident angle cannot be calculated through the deflection angle of AOTF corresponding to the two intersections. When the frequency is selected as $f_{2}$, the tuning curve cannot be accurately obtained due to the error in the actual design parameters, and the approximate value will also produce an error. When the frequency is selected as $f_{3}$, the extreme point of the curve corresponds to the highest diffraction efficiency under the premise of momentum mismatch, requiring no accurate tuning curve relationship. In practice, the actual frequency is 0.5 MHz lower than the frequency calculated by the ideal tuning curve. Finally, the incident polar angle corresponding to the highest diffraction efficiency point is the desired one. The experimental setup is the same as the method above. Figure 8b shows the measured data and the fitting results.

Third-order polynomial fitting is performed based on the measured data. The diffraction efficiency corresponding to the extreme point of the fitting curve is 59.91%, and the incident polar angle corresponding to the extreme point is 0.52°. Compared with the minimum-central wavelength test result, there is an error of 0.03°. The results obtained by the two test methods are very close, indicating the feasibility and accuracy of the two. Finally, the AOTF design parameters obtained from the test are taken into the model to calculate the tuning curve. Before correction, $\alpha =8.9^{\circ }$, $\theta_{i}=20.48^{\circ }$. After correction, $\alpha =8.9^{\circ }$, $\theta_{i}=20.70^{\circ }$. The geometric layout in Figure 1 has changed, so the tuning curve (model calculation) has changed accordingly. The measured results of the tuning curve and the results of model calculation before and after parameter correction are shown in Figure 9a. The results show that the theoretical tuning curve can be more consistent with the measured data only after GPC is corrected. The relative difference in central frequency can be expressed by Equation (9). The results are shown in Figure 9b.
(9)$$relative\;error=\frac{\left|f_{c}-f_{t}\right|}{f_{t}}\times 100\%,$$
where $f_{c}$ is the theoretical value of the central frequency. $f_{t}$ is the measured value of the central frequency. Within the test band, the frequency error can be controlled within 1% after the geometric parameter correction of the AOTF crystal.

Then, we performed five complete measurements with the two different methods. The uncertainty was calculated by Equation (10).
(10)$$\{\begin{array}{l} U_{A}\approx \delta_{x}=\sqrt{\sum_{i=1}^{n}{\left(x_{i}-\bar{x}\right)}^{2}/(n-1)}\\ U_{B}\approx \Delta ins\\ U=\sqrt{U_{A}{}^{2}+U_{B}{}^{2}} \end{array}$$
where $U_{A}$ is the measurement uncertainty; $\delta_{x}$ is the standard deviation of the measurement result of the vertical incidence angle ($\theta_{i}$); Δins is the uncertainty of the measuring instruments. It is assumed that $\Delta ins=0.01^{\circ }$ to maintain the consistency of the two methods; U is the combined uncertainty. The calculation results are shown in Table 2. The advantages and disadvantages of the two methods are also given.

From Table 2, the time required for method A is much longer than for method B. In our case, method A is more precise because the transducer will produce a power fluctuation during frequency adjustment. Method B is greatly affected by the experimental conditions. Besides the above, the measured values of error in the tuning curves of both methods fall within the confidence intervals.

Finally, we carried out the mid-infrared spectral imaging experiment, and the experimental schematic diagram is shown in Figure 10.

As seen in Figure 10, the distance between the target and AOTF spectrometer was far in the experimental setting, so the pre-collimating mirror group (telescope) could be omitted. It was only necessary to measure the aberration of the imaging mirror group and add the aberration parameters to the simulation model. The temperature of the heating plate could reach 100 °C. The diameter of the dots was 6 mm, and the distance between the calibration points was 20 mm. The coordinates of the calibration points were obtained by the method of geometric center extraction, and the unsatisfactory calibration points were eliminated. The size of the detector’s target plane was 256 × 320 and the pixel size was 30 μm. The focal length of the imaging mirror group was 50 mm. The ultrasonic cut angle provided by the manufacturer was 8.9°. The luminous aperture of AOTF was 20 × 20 mm. The length of the piezotransducer was 21 mm. Different GPCs were brought into the model to calculate the theoretical imaging points, and the results are shown in Figure 11 without considering the alignment errors.

In Figure 11, the circles represent the simulated imaging points, and the red asterisks represent the actual imaging points. If we remove the two calibration points within the red dotted box, we can find that the simulated imaging points are all located near the actual imaging areas. After the geometric calibration of the AOTF crystal, the ideal imaging points are closer to the actual imaging points, but the effect is not obvious. The main reasons lie in the following aspects: (1) the aberration of mid-infrared AOTF is much smaller than that of the visible band; (2) the geometric calibration result of AOTF crystal has little difference from the reference value; (3) limited by the angular aperture, the field of view of the system is ±3°. In this angle range, the aberration is small. The $error_{RMS}$ can be calculated by Equation (11).
(11)$$error_{RMS}=\sqrt{\frac{1}{N}\sum_{j=1}^{N}\left[{\left(x_{actual,j}-x_{model,j}\right)}^{2}+{\left(y_{actual,j}-y_{model,j}\right)}^{2}\right]}.$$
where $x_{actual,j}$ is the row coordinate of the diffracted light imaging position. $x_{model,j}$ is the row coordinate of the calculated imaging position. y is the column coordinate. N is the total number of sampling points. Before the correction, $error_{RMS}=2.01$. After correction, $error_{RMS}=1.07$. The pixel difference for each point is shown in Figure 12. The fitting curve adopts cubic polynomial fitting. The pixel difference is controlled at approximately one pixel after correction.

## 5. Conclusions and Future Directions

Before designing and manufacturing a spectrometer based on AOTF, it is necessary to theoretically analyze the key parameters, such as the spectral bandwidth, imaging distortion, and so on. If the GPC cannot be obtained accurately, it will be impossible to conduct simulation analysis on the premise of accuracy, making the results meaningless. For the AOTF devices not strictly conforming to the principle of parallel tangent, two test methods to determine the design parameters are proposed, and the final test results of the two are highly consistent, with an error of 0.03°. Finally, the accuracy of the obtained design parameters is verified by the tuning curve and geometric calibration experiment. Results reveal that the accuracy of GPC is the premise on which the AOTF model can be used for ray tracing or analyzing the response of the device or the system. It is also the original intention of this work.

As for the advantages and disadvantages of the two methods, we believe that the data measured by the minimum-central wavelength method will have better stability. Because of the heating due to the operation of AOTF, the fluctuation in diffraction efficiency is influenced more seriously than the shift in the central wavelength.

The design of all AOTFs is based on the optical anisotropy of crystals; until now, however, only uniaxial crystals have been used. The method in this paper is also only applicable to the calibration of the geometric parameters of uniaxial crystals. The spectral response of AOTF based on biaxial crystal is more complex, and the calibration method remains to be determined.

## Figures and Tables

**Figure 1 materials-16-02341-f001:**
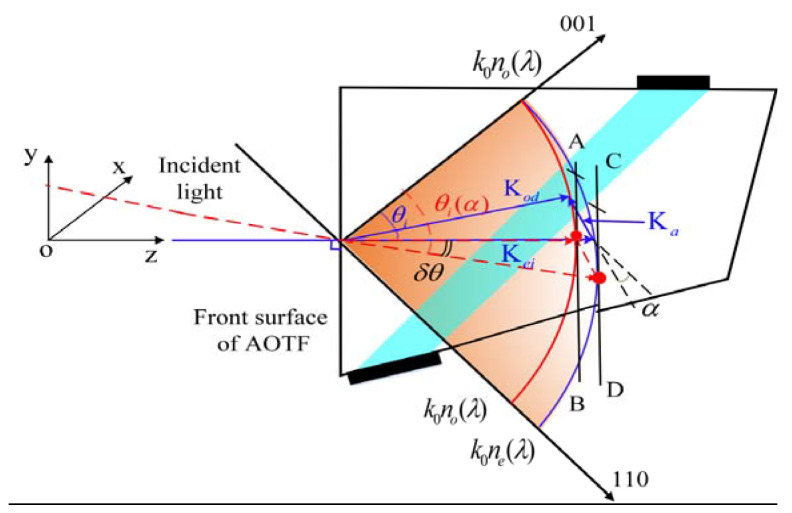
Wave vector layout under the parallel tangent principle (e→o mode). $K_{ei}$, incident light is extraordinary; $K_{od}$, diffracted light is ordinary; $K_{a}$, acoustic vector; **oy**, the direction of acousto-optic diffraction.

**Figure 2 materials-16-02341-f002:**
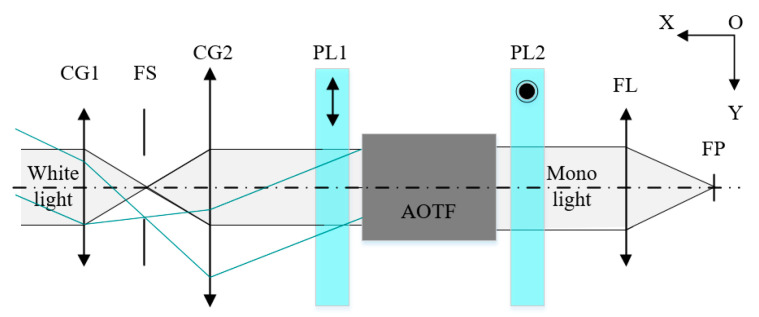
The simulation optical scheme of the system. CG1 and CG2: collimator group; FS: field stop; PL1 and PL2: polarizer; FL: focal lens; FP: focal plane; OY: the direction of acousto-optic diffraction and the column direction.

**Figure 3 materials-16-02341-f003:**
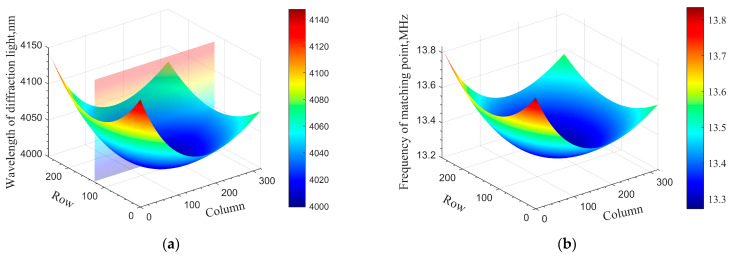
Mid-infrared AOTF image plane drift. (**a**) Central wavelength drift and (**b**) central frequency drift.

**Figure 4 materials-16-02341-f004:**
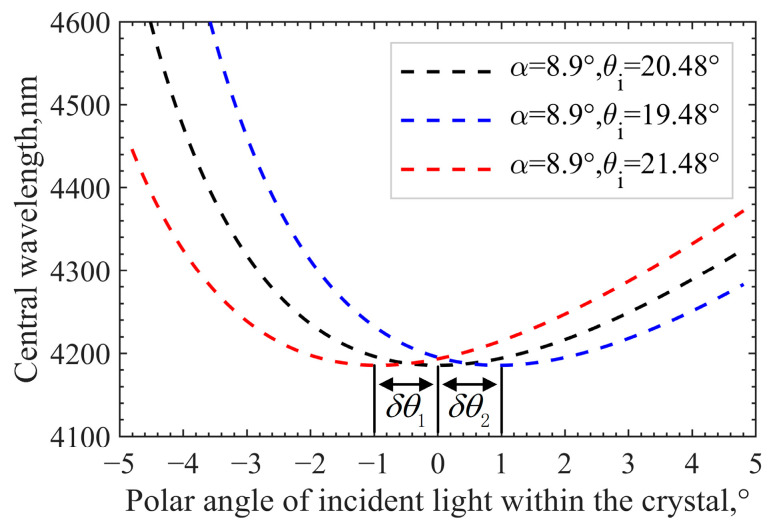
Influence of the GPC on spectral response.

**Figure 5 materials-16-02341-f005:**
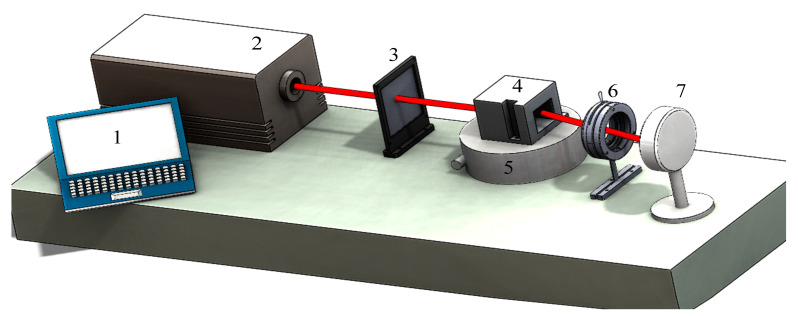
The schematic diagram of AOTF geometric parameter calibration. 1—computer; 2—mid-infrared laser; 3—polarizer; 4—AOTF; 5—precision turntable; 6—diaphragm; 7—optical power meter.

**Figure 6 materials-16-02341-f006:**
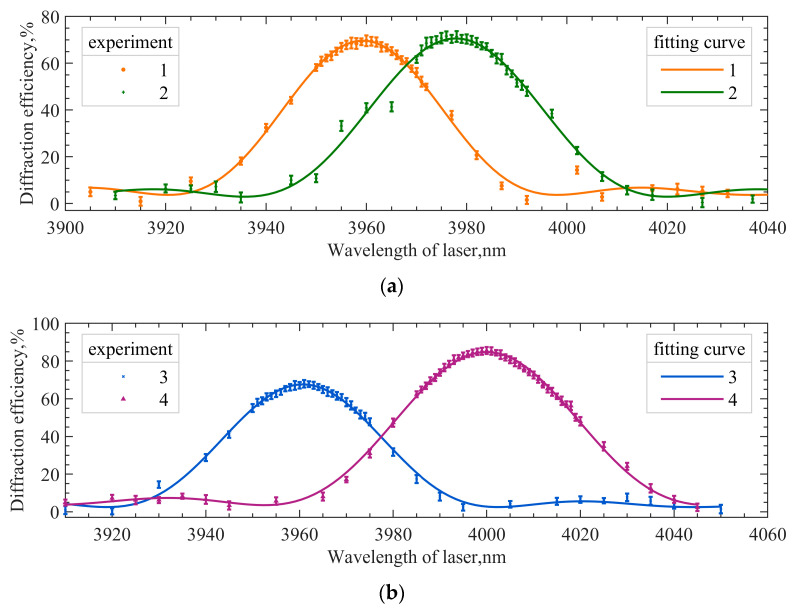
Spectral responses at some different incident angles. (**a**) 1: $\theta_{p}=0^{\circ }$; 2: $\theta_{p}=-3^{\circ }$; (**b**) 3: $\theta_{p}=-1^{\circ }$; 4: $\theta_{p}=-4^{\circ }$.

**Figure 7 materials-16-02341-f007:**
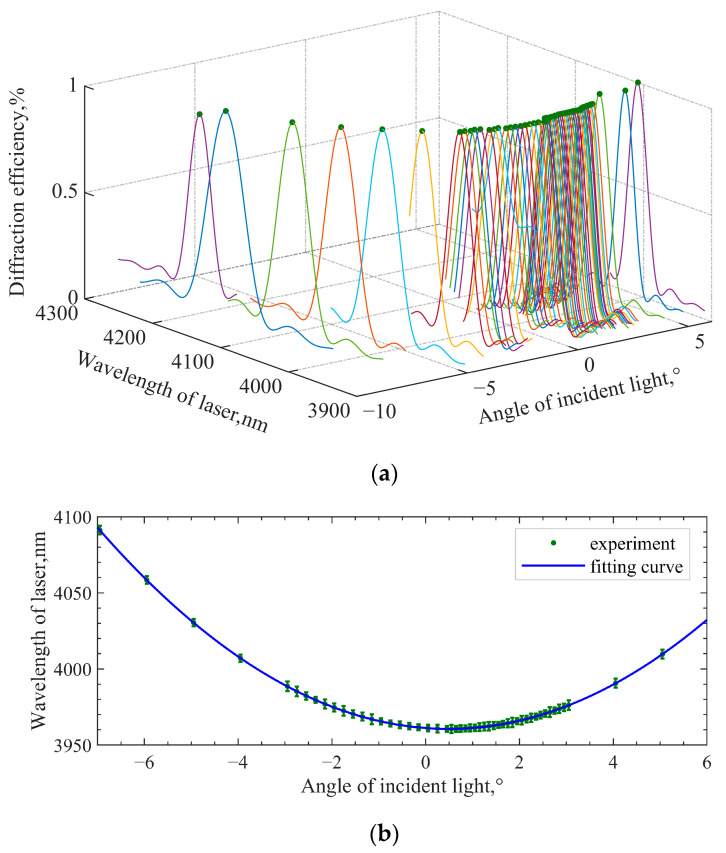
Measurement and fitting results of the minimum-central wavelength method. (**a**) Three-dimensional diagram of spectral responses at different angles; (**b**) fitted curve of central wavelengths.

**Figure 8 materials-16-02341-f008:**
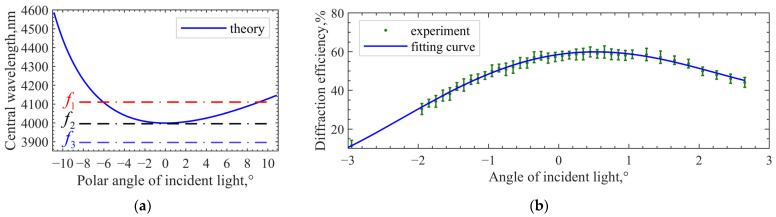
The minimum-frequency method. (**a**) Method of frequency setting; (**b**) measured data and fitting results.

**Figure 9 materials-16-02341-f009:**
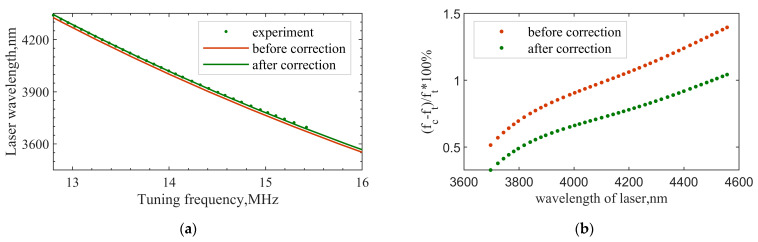
Correction result of the tuning curve. (**a**) Correction of tuning curve; (**b**) relative difference in central frequency.

**Figure 10 materials-16-02341-f010:**
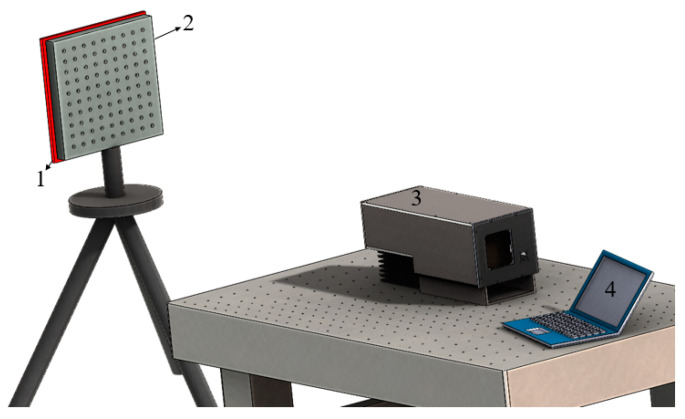
Schematic diagram of mid-infrared AOTF spectral imaging experiment for geometric calibration. 1—heating plate; 2—geometric calibration target; 3—AOTF spectrometer; 4—computer.

**Figure 11 materials-16-02341-f011:**
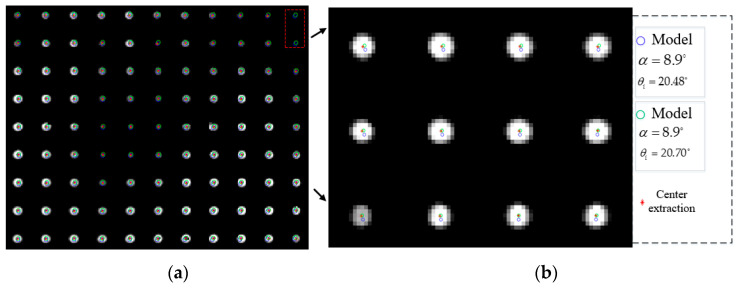
Spectral imaging experiment based on mid-infrared AOTF. (**a**) Actual image of diffracted light; (**b**) the partially enlarged view of the target.

**Figure 12 materials-16-02341-f012:**
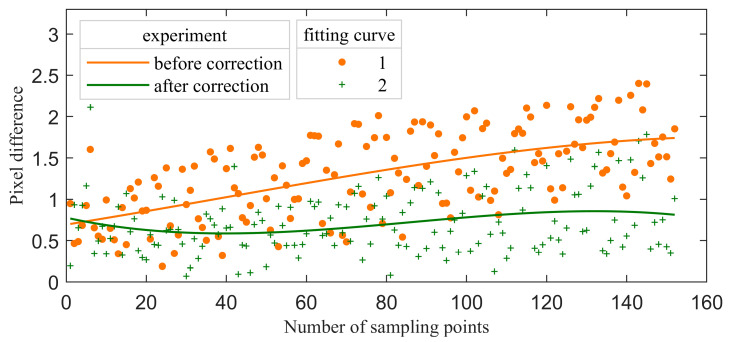
The pixel difference for each point.

**Table 1 materials-16-02341-t001:** The works related to AOTF in recent years.

AOTF-Related Work	Recent Research
Calibration	Bürmen, Li, Vila-Francés, spectral calibration [21,22]; Pozhar, Machikhin [23,24,25] and Liu [26], geometric calibration; Katrašnik, Healey, Tian, radiometric calibration [30,31,32]; Trutna, Yano, thermal calibration [33,34]; Shi [27], 3D calibration; Katrašnik [28,29], automatic geometrical calibration methods.
Optical design and optimization	Batshev and Machikhin [17], typical optical schemes; Batshev and Gorevoy [18], Zemax integration.
Operating mode	Zhang [13], noncollinear AOTF operating mode, the principle of parallel tangent, tuning curve.
Ray tracing	Yushkov [15], tuning frequency at oblique incidence; Pozhar and Machikhin [23], 2D to 3D ray tracing.

**Table 2 materials-16-02341-t002:** The advantages and disadvantages of the two calibration methods.

Method	Calibration Time/Min	Calibration Results/°	Frequency Error
A: The minimum-central wavelength method	~270	α=8.9 $,\theta_{i}=20.70\pm 0.04$	≤1%
B: The minimum-frequency method	~30	α=8.9 $,\theta_{i}=20.72\pm 0.10$	≤1.05%

## Data Availability

Not applicable.

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
