# Peer review of "The Calibration Methods of Geometric Parameters of Crystal for Mid-Infrared Acousto-Optic Tunable Filter-Based Imaging Systems Design"

_materials, 2023, doi:10.3390/ma16062341_

Round 1

Reviewer 1 Report

The manuscript ID materials-2251943 mainly presents a study about two calibration methods based on the principle of parallel tangent. The study is focused on the minimum-central wavelength method and the minimum-frequency method. The results are interesting but some issues are present in the report. Here are some points to the authors:

1.   From the introduction is not clear how the authors proposed the two methods studied. Additional descriptions justified the importance of the research are kindly requested.

2.   How is the influence of inhomogeneity of the crystal in the main findings?

3.   Please comment about the influence of incident polarization in the measurements.

4.   It would be interesting if the authors could extend some perspectives for the assistance of this research in other acousto-optics techniques. The authors are invited to see for instance studies involving plasmonic sensing doi:10.3390/mi8110321.

5.   Advantages and disadvantages of the two methods proposed in respect to comparative works should be critically discussed. You can consider for instance https://doi.org/10.3390/ma14133454

6.   The color bar in figures 3 and 5 is missing.

7.   Some equations require citation.

8.   Please add two keywords to increase the probability that this work can be selected in a search by an internet tool in case it is published.

9.   The error bar in experimental data should be mandatory.

10.               The parameters in figure 1 must be described in the text close to the figure.

Reviewer 2 Report

The paper presents an interesting calibration procedure.

Considering the topic, I expected a much more detailed description of uncertainty of the procedure.

Authors refers to accuracy somewere (the correct term should be uncertainty) but the dscusson and the sources of the numbers provided are too fuzzy and not sound.

Reviewer 3 Report

Before evaluating the proposed manuscript sent to the journal Materials, the authors need to redo the text. First of all, it is necessary to explain the origin of equations 1-3. What is the physical meaning of these equations and what does plus minus mean in the equation 1! The text in chapter 2 should be carefully reworked in order to eliminate such mistakes as “To meet the requirement of the momentum matching condition”,  and “In addition to the momentum matching condition, the AOTF imaging spectrometer needs to meet the requirement of the highest throughput. ” Which means  on the Fig.1!

In lines 151-152, the authors wrote that “As shown in Equation (4) and Equation (5). ” These equations don't tell the reader anything! Lines 155 to 158 are a set of incomprehensible sentences! What is the origin of equations 6 and 7! Then suddenly some error is declared in 0.1 %!

Let the authors take as an example any article from Phys. Rev. or from J. Chem. Phys. and they will see how to write scientific articles!!

Reviewer 4 Report

This work proposes two calibration methods based on the principle of a parallel tangent: the minimum-central wavelength method and the minimum-frequency method. The paper's contribution to existing knowledge in this research field is well justified. The authors mentioned some recent techniques, but the paper needs to address the motivation for developing another method. The paper needs to contribute more, and the following points can improve the manuscript.

1.     The title can be improved. The term AOTF should be defined before using it.

2.     Enhance the abstract and introduction to show the motivation for this work.

3.     A comparative study can be added to a related work section in table form to show the recent efforts.

4.     Figures 6, 7, 8, 9, 10, and 11 should be improved.

5.     Double-check all the equations to be true.

6.     The proposed method should be compared with recent techniques.

7.     There should be some discussion on the limitations of the method presented in a separate section.

8.     The manuscript organization should be improved. 

9.     Improve the English of the work. There are too many problems with paper typesetting.

10.  The paper is unsuitable for acceptance in its current form. The article needs rewriting to address the comments mentioned above. 

Round 2

Reviewer 1 Report

The reviewed manuscript is clear and it can be useful for future research in the topic of mid-infrared acousto-optic tunable filter based imaging systems design. The authors have successfully addressed most of the points raised in the review stage, and in my opinion, this work can be considered for publication in present form.

Reviewer 3 Report

In this paper, a method for calibrating the geometric parameters of a crystal for designing visualization systems based on acousto-optical tunable filters (AOTF) of the mid-infrared range is proposed. It is shown that AOTF calibration is a complex topic with many different aspects. As for geometric calibration, the author showed that it includes not only processing errors and error correction in the optical system, but also the error of the geometric parameters of the crystal (GPC). GPC is a preset input signal in the optical design and optimization of Zemax, which determines key parameters, including spatial resolution, field of view and aberration. In particular, aberration compensation in optical design requires precise GPC values. The author has proposed two calibration methods based on the principle of parallel tangent: the first, the method of minimum central wavelength, while the second method is the method of minimum frequency. It is shown that the deviation of the angle of incidence of the parallel tangent, calibrated by two methods, is 0.03. As a result, it is shown that the tuning curve calculated theoretically using calibrated geometric parameters of AOTF is consistent with the tuning curve measured in practice. The results show that GPS accuracy is a prerequisite on which the AOTF model can be used to trace rays or analyze the response of a device or system.

The work undoubtedly deserves to be published in the journal Materials.

Reviewer 4 Report

The authors have addressed most of my concerns. The paper can be accepted.